

# Large-scale vegetation responses to terrestrial moisture storage changes

Robert L. Andrew[1], Huade Guan[1,2] and Okke Batelaan[1,2]

[1]School of the Environment, Flinders University, Bedford Park, 5042, Australia
[2]National Centre for Groundwater Research and Training, Bedford Park, 5042, Australia

*Correspondence to*: Robert L. Andrew (robert.andrew@flinders.edu.au)

**Abstract**

The Normalised Difference Vegetation Index (NDVI) is a useful tool for studying vegetation activity and ecosystem performance at a large spatial scale. In this study we use the Gravity Recovery and Climate Experiment (GRACE) total
water storage (TWS) estimates to examine temporal variability of NDVI across Australia. We aim to demonstrate a new method that reveals the moisture dependence of vegetation cover at different temporal resolutions. Time series of monthly GRACE TWS anomalies are decomposed into different temporal frequencies using a discrete wavelet transform and analysed against time series of NDVI anomalies in a stepwise regression. Results show that combinations of different frequencies of decomposed GRACE TWS data explain NDVI temporal variations better than raw GRACE TWS alone.
Generally, NDVI appears to be more sensitive to inter-annual changes in water storage than shorter changes, though grassland-dominated areas are sensitive to higher frequencies of water storage changes. Different types of vegetation, defined by areas of land use type show distinct differences in how they respond to the changes in water storage which is generally consistent with our physical understanding. This unique method provides useful insight into how NDVI is affected by changes in water storage at different temporal scales across land use types.


**Keywords:** Vegetation index, NDVI, GRACE, ecosystem performance, water storage, wavelet analysis, regression analysis, land use type

## 1. Introduction

In many parts of the world, such as Australia, water storage is the dominant limiting factor in vegetation growth (Donohue et al., 2008). As such, changes in water storage can lead to changes in vegetation mass and greenness (Yang et al., 2014). As vegetation plays a vital role in gross primary production and the carbon and hydrological cycles, studies of the temporal and spatial variation of vegetation are vital for understanding ecosystem performance and its climatic responses (Campos et al., 2013). As the climate and water resources change as a result of natural and anthropogenic influences, understanding how
fluctuations in water storage is associated with biomass changes can have profound importance in the future.

Previous studies have used different hydrological parameters to examine the effect of hydrological changes on ecosystem performance. Most commonly, precipitation and soil moisture have been used as defining variables (Chen et al., 2014, Huxman, 2004, Méndez-Barroso et al, 2009, Wang et al., 2007). Both of these have shown generally meaningful
correlations with ecosystem performance (by various measures such as Normalised Difference Vegetation Index (NDVI) and above-ground net primary production). However, both indicators have shown limitations. The total amount of precipitation is





not necessarily used by vegetation in an ecosystem. Part of precipitation is lost from the ecosystem as runoff or soil

evaporation (Liping et al., 1994). Only the part which is retained as soil moisture in the root zone can be viably consumed by

vegetation, categorised as 'effective precipitation' (Bos et al., 2009). For a given amount of rainfall the fraction of effective

precipitation varies spatially due to differing geographical features, soil types, and vegetation cover conditions. Soil moisture

gives a better representation of the water that becomes available to plants. However, in situ soil moisture data is generally

limited and spatially (vertically and horizontally) sparse. Estimations from land surface models are often highly uncertain

(Chen et al., 2013).

More recently Yang et al. (2014) used monthly total water storage anomalies (TWS*) from the Gravity Recovery and

Climate Experiment (GRACE) to examine hydrological controls on variability in surface vegetation. GRACE provides

monthly global terrestrial water storage derived from variations in the earth's gravity field. The authors suggested that where

large surface water reservoirs do not exist, GRACE TWS changes are mostly from soil moisture and groundwater, making it

ideal for examining hydrological controls on vegetation activity. GRACE is found to be a good indicator of seasonal

variability in surface greenness over mainland Australia (Yang et al., 2014). For the period 2003-2010, for which GRACE

data is available, changes in NDVI* are explained more strongly by GRACE TWS* than by precipitation, suggesting it

poses a more direct influence on surface greenness and ecosystem performance.

GRACE TWS gives the total relative water storage per 100 km by 100 km cell. This is the sum of surface water, soil water,

groundwater, ice etc. We previously developed an approach to 'split' GRACE TWS into shallow and deep subsurface

storage components using discrete wavelet decomposition (Andrew et al., 2016). In this study, we aim to expand on the

general findings of Yang et al. (2014) by decomposing GRACE TWS* into different temporal components and analysing

them against NDVI*. Given that root zone water storage is the source of water to vegetation we hypothesize that

decomposed TWS* data that reflects the temporal patterns of the root zone will perform better than the total TWS* in

association with NDVI*.


The questions we seek to address are (1) does the decomposed TWS* data show a better relationship to NDVI* than the

'raw' TWS* data; (2) how does the sensitivity of NDVI* in response to changes in TWS* vary spatially; and (3) which

temporal components of TWS* are most significant in influencing NDVI* for different land use types across Australia.

## 2. Data

### 2.1 GRACE data

We use GRACE total water storage (TWS) data from The University of Texas Centre for Space Research (CSR), and

NASA's Jet Propulsion Laboratory (JPL). The GRACE data sets are freely downloaded from the GRACE Tellus website



(http://grace.jpl.nasa.gov/data/get-data/). Data is suitably post-processed, including applying the recommended scaling

correction (Swenson and Wahr, 2006). The scaling coefficients are in part designed to remove leakage errors (Landerer and

Swenson, 2012). Monthly data from March 2003 to December 2014 is used. The average of the two data sets is calculated

for each cell at each month to reduce the uncertainty. The data is presented spatially in 100 km by 100 km grid cells. We

selected which cells should be included based on a shape file of Australia. If at least two thirds of the cell is part of the

continent they are included, this eliminates some cells which covered only a small coastal land mass.

There are a few occurrences of missing data in the GRACE data set. These months of missing data are filled with a simple

temporal interpolation using the months either side. Because of the monthly temporal resolution this is deemed appropriate

and maintains the average seasonal cycle well (Long et al., 2015).

Climatological anomalies of GRACE TWS are used in order to remove seasonality in the data which would otherwise result

in large, but irrelevant and misleading correlations between variables examined in this study. The anomalies are calculated

following the method of Yang et al. (2014), as shown in Eq. (1).

$$X^*(i,j) \;=\; X(i,j) - \frac{1}{n} \sum_{j=1}^{n} X(i,j) \tag{1}$$

where $X^*$ represents the climatological anomaly of X (i.e. raw GRACE TWS), $i$ is the month, $j$ is the year and $n$ is the total

number of years.

New lagged GRACE TWS* anomaly data sets are produced by offsetting the GRACE data from the NDVI data by one to

six months. This is to allow any delays in NDVI response to water storage to be revealed (Farrar et al., 1994) and is

discussed in more detail in the methodology section.

20         **2.2 NDVI data**

We use GIMMS 3g NDVI data for the same time period as the GRACE data. The data is downloaded from the NASA

database. The NDVI data is produced at a smaller spatial scale (.25 by .25 degrees) than GRACE. They are rescaled to match

the GRACE cell size using the resampling tool in ArcGIS. Like the GRACE data, only cells which contain at least two thirds

land are used, and missing data are filled by a temporal interpolation. Similar to the GRACE data also for NDVI

climatological anomalies are calculated using equation 1.

         **2.3 Land Use Type data**

The moderate-resolution imaging spectroradiometer (MODIS) land use data is used to identify different land use types

across Australia. It is freely available online from http://glcf.umd.edu/data/lc/. In regards to rescaling and cell selection, the

same procedures are applied as in the case of NDVI data. In Australia, MODIS land use type data defines 12 different





classes of land use. This is reduced to five (or six including barren land) classes by grouping similar classes such as different

types of forests. The resulting land use types are: forest, shrubland, savanna, grassland, and agricultural land (Table 1).

Figure 1 shows the spatial distribution of different land use types across Australia, grouped as previously stated (Table 1).

Note no analysis is performed for areas considered barren, due to a lack of vegetation.

### 3  Methodology

#### 3.1  Wavelet decomposition

GRACE TWS* is decomposed into different signals using a discrete wavelet transform. Introduced in the early 1980s, a

wavelet is a mathematical function used to divide data series into different-frequency components (Goupillaud et al., 1984).

The method expresses decompositions as a multitude of smaller 'waves' at different frequencies (He et al., 2013). The

Meyer wavelet is applied here to decompose GRACE TWS* into components at different temporal scales and is suitable for

this temporal data (He & Guan, 2013). This is relatively easy to achieve by means of a simple MATLAB code using the

'wavdec' function. Data is decomposed into 'approximation' and 'detail' components, each representing a different temporal

scale. Approximation series maintain trends in the data while detail series neglect trends (Nalley et al., 2012). The resulting

time series are labelled A1, A2, A3, A4 and D1, D2, D3 D4 for approximations and details respectively, with the time scale

increasing with the decomposition number e.g. A1/D1 (2-month scale), A2/D2 (4-month scale), A3/D3 (8-month scale) and

A4/D4 (16-month scale). Four levels can be reasonably extracted given the data length and monthly frequency of the data.

Further decomposition would result in roughly 3 and 6 year time scales which is too coarse for a time series of only 11 years

of raw data. Because all but the lowest approximation levels contribute partly to details, we only use the lowest frequency

approximation, along with all of the details. The sum of these (D1, D2, D3, D4, A4) equals the raw signal (Fig. 2). So, five

wavelet decomposition series are produced for GRACE data as well as each of the six lagged series' for each decomposition

level giving a total of 35 water storage time series.

#### 3.2  Stepwise regression

We used a stepwise regression for every cell with NDVI* as the dependant variable and the GRACE TWS* decompositions

as predictor variables. Given the time series of the data, 35 predictor variables is too many for a stepwise regression to

function properly. The stepwise regression is run multiple times and the best predictor variables are chosen narrowing them

down to nine. The choice is made based on the amount of cells selected for each variable from the stepwise regression and

how relevant they are given their spatial coherence. In general, the predictor variables excluded from the stepwise regression

are not included in any cells across the country. The remaining variables are (subscript denotes lag in months) $D1_0$, $D2_0$, $D3_0$,

$D3_1$, $D3_2$, $D4_0$, $D4_1$, $A4_0$, and $A4_6$.





### 4. Results

As a proof of concept, the relationships between raw GRACE TWS* and NDVI*, and decomposed GRACE TWS* and

NDVI* are compared (Fig. 3). The results for the decomposed TWS* data are based on a selection of decomposed time

series selected by the stepwise regression. For each cell the correlation coefficient between NDVI* and the regression

estimates ( $r$ ) is calculated. In order for the tests to be comparable, lagged data is not included in the decomposed TWS

dataset for this demonstration, it shows purely how decomposed data improves the relationship. A scatter of the r values

shows a clear improvement in the relationship when decomposed GRACE TWS* data is used as opposed to raw, with all

points above the 1:1 line. Student-t tests confirmed that the stepwise regression results are statistically highly significant with

a t-statistic p value of respectively 2.3 and .00014.


Lagged data ensures the relationship between NDVI* and TWS* is well represented, but the decomposed frequency of the

TWS* data is the focus in this study. Though the stepwise regression is performed using nine variables including lags where

suitable, the results herein are presented as only five variables, D1, D2, $D3_L$, $D4_L$ and $A4_L$. For each detail or approximation

level using different lags, one variable is created by combining the results of different lagged data sets together to present the

results i.e. $D3_L = D3_0+D3_1+D3_2$.

It is important to recognise how the variables that are included in the stepwise regression vary spatially to understand how

vegetation responds to different temporal patterns water storage across the continent. For a variable to be included in the

stepwise regression it does not have to show a positive correlation. Figure 4 shows which variables are included in the

regression for each cell across Australia. Where no lagged data is used (D1 and D2) the colour denotes whether the

coefficient is positive or negative. Where lagged data is used ($D3_L$, $D4_L$ and $A4_L$) the colour denotes whether all coefficients

for a cell had the same -/+ sign or not. Figure 4 shows that while $A4_L$ is included across most of the country, one of the

lagged data sets, $A4_6$ has a large amount of negative coefficients included in the regression (see appendix 1). A possible

explanation for this is that NDVI is susceptible to the 'memory effect', where past inputs and outputs affect responses in the

system (Shook & Pomeroy, 2011).





Overall, the number of cells covered by each different decomposition level increases as the decomposition time scale increases. This shows that in general, NDVI changes pertain to longer time-scale water storage changes and is not affected as much by changes on monthly time scale.

While understanding which variables are used in each cell is important, it is more important to know their relative impact on NDVI*. The relative weight of each variable is calculated to show the importance of each on vegetation in different land use types. Of the included variables in each cell, the relative weight of each variable is calculated using Eq. (2).

$$W = \frac{(C \cdot \sigma_X)}{\sigma_{NDVI}} \tag{2}$$

Where $W$ is the relative weight, $C$ is the coefficient, $\sigma_X$ is the standard deviation of the decomposed data anomaly ($X$), $\sigma_{NDVI}$ is the standard deviation of the NDVI anomaly. Figure 5 shows which variable has the highest relative weight in each cell. $A4_L$ is the dominant variable, covering the majority of the country, and is a low frequency trended signal. D2, a higher frequency signal is the second most dominant variable and shows generally clear spatial coherence.

The relative weights for all cells of each land use type are combined and presented as a relative weight percentage per land use type (Fig. 6).

Forested areas have only low frequency decompositions included, with $A4_L$ being the most dominant. This is expected as forests have deep root systems which tap into water stores which change slower than shallower water (Backer et al., 2003).

Therefore, their water availability is less likely to be affected by short-term rainfall or evaporation, relying more on long term hydrological trends. Shrubland, savanna and grassland show nearly identical distributions of weights. Grassland shows a marginally higher percentage of the D1 and D2 variables, which is consistent with our physical understanding as they are fed by shallow soil moisture which varies at a short time frequencies. While all are defined differently, the three land use types have overlapping characteristics, most commonly the widespread presence of short grasses (Friedl et al., 2002) and

shallow root systems. These short grasses respond to changes in the shallow top layer of the soil which is influenced at high temporal frequencies by rainfall events and evaporation. The similarity in the result of these three land use types suggests that they are hardly distinguishable by GRACE, likely due to the spatial extend of GRACE cells. For example, where sparse trees exist in a savanna, their influence on the shallow soil moisture may be negligible compared to the large coverage of grasses, thus showing a very similar pattern to grassland.


### 5. Discussion



Using wavelet decomposed GRACE TWS* data proved to improve the correlation between water storage and NDVI*. A previous study by Yang et al. (2014) showed that GRACE is a superior indicator of surface greenness than soil moisture or precipitation, which were earlier used as indicators (Chen et al., 2014, Huxman, 2004). Temporal decomposition of GRACE TWS* produces a new temporal dimension that allows the data to be analysed to its full potential. As demonstrated in Fig. 3,

the decomposed TWS* data is the better associated with the surface greenness than the raw TWS*. Furthermore a better understanding of how surface greenness changes with water storage spatially and temporally is achieved, with different levels of decomposition existing in spatial clusters across the country. The dominance of $A4_L$ as the most highly weighted predictor variable indicates that generally vegetation responds to low frequency (inter-annual) changes in water storage across Australia.

An interesting result is the large amount of negative coefficients produced from the stepwise regression for $A4_6$. Two possible explanations exist. A 6-month lag may correspond to the opposite seasons e.g. wet 6 months ago, dry now, potentially serving as an indicator of water storage potential. Alternatively, vegetative systems may be susceptible to the 'memory effect.' Specifically, this would suggest that for most of the continent, trends at the A4 scale (roughly annual)

influences vegetation responses to water storage changes six months later in these areas. Such a memory effect can serve as an indicator of an ecosystem's capacity to store water, as well as carbon and nitrogen (Schwinning et al., 2004).

The weight distribution of different decompositions across land use types generally matches our physical understanding. Note firstly that all five land use types have $A4_L$ as a large component of their total weight. This is a further indication of the

general response of vegetation to low frequency changes in water storage. Forested areas are only composed of $A4_L$, $D4_L$ and $D3_L$, irrelevant to high frequency changes in water storage. This matches our physical understanding as forests have deeper root systems which rely on seasonal changes or long term hydrological trends. Interestingly, shrublands, grasslands and savannas show a near identical composition of relatively weighted decompositions, with grasslands showing a slightly higher weight percentage of D1 and D2. The three land use types are all grass dominated, with the addition of sparse trees and

shrubs in savannas and shrublands. As the resolution of GRACE cannot pick up these additions, it is possible that they all appear as grassland, or at least skewed that way, as that is the dominant vegetation cover. The dominance of D1 and D2 across these land use types is typical of relatively dynamic, grass dominated regions.

The combination of weights that make up the total for agricultural land is less straightforward. D2 and $A4_L$ contribute to

large portions of the total. One major difference between agricultural land and the other land use types is the anthropogenic contributions to the land, including the additions of livestock grazing (Yates et al., 2000). The other land use types are



generally self-sufficient/limiting at the cell scale, so the interruption of the natural cycle of the vegetation in agricultural areas is a potential anomaly, disturbing any predictable composition of relative weights.

Our method of using decomposed terrestrial water storage as an improved indicator of surface greenness has potential environmental benefits. It allows for an improved understanding of how vegetation responds to changes in water storage at a spatiotemporal level. This in turn serves as a better indicator of ecosystem performance and carbon fluxes. With predictions of terrestrial water storages to decline in the future (Gleick, 1989), the method could be highly useful for predicting carbon fluxes and ecosystem performance based on future water storage estimates. Furthermore, the global mapping of GRACE and NDVI (as well as other vegetation indexes) means that it could be applied globally.

### 6. Conclusion

In this study we aimed to increase the understanding of the links between GRACE TWS* and NDVI* by using a decomposed TWS* data. Combinations of decomposed GRACE TWS* data show an improved relationship with NDVI* than raw GRACE TWS* data alone. Varying decomposed frequencies show spatial coherence in parts of the country,

sometimes independently and sometimes overlapping other decomposed frequencies. Generally, NDVI is influenced by low frequency changes in water storage, however there are some areas which are also sensitive to high frequency changes. NDVI is susceptible to a memory effect which depends on previous TWS conditions, 6 months generally. The total influence of NDVI changes is made up of storage changes over different time periods. These vary depending on the land use type and the results are aligned with our physical understanding. This analysis could be used further to continue to improve our

understanding of vegetative responses to storage change in Australia and globally and benefit predictions of ecosystem performance and carbon fluxes.

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




5 **Figures**

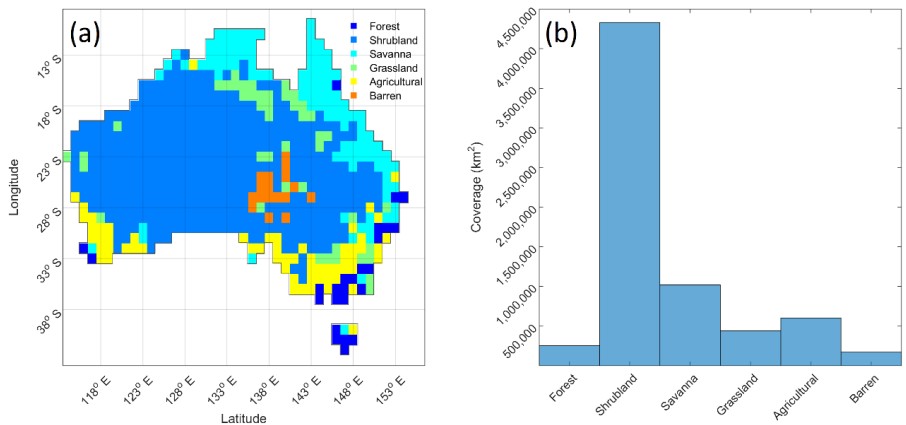

**Figure 1:** (a) The spatial distribution of various land use types across Australia and (b) the area covered by each land use type.

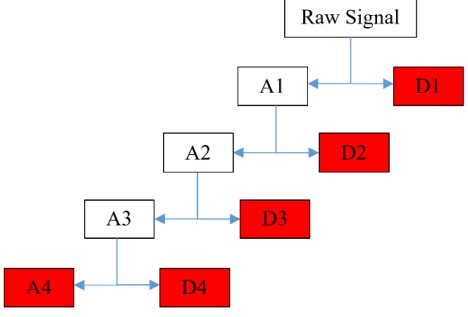

**Figure 2:** The structure of a wavelet decomposition; decomposition levels used in this study are highlighted in red.

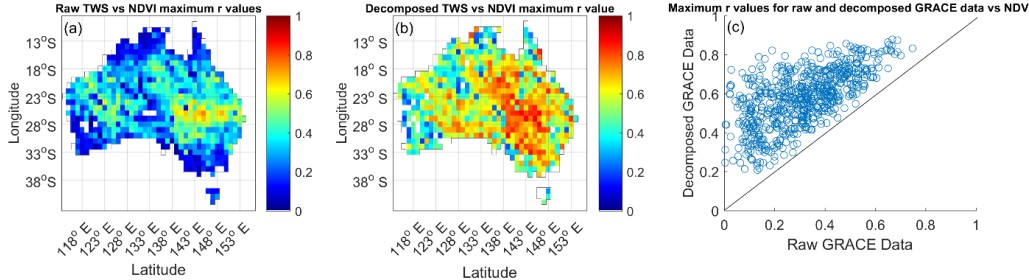



**Figure 3:** (a) The *r* values using raw TWS* and NDVI*. (b)The r values using decomposed TWS* and NDVI*. (c) a scatter of the results shows a clear improvement in the relationship when decomposed data is used.

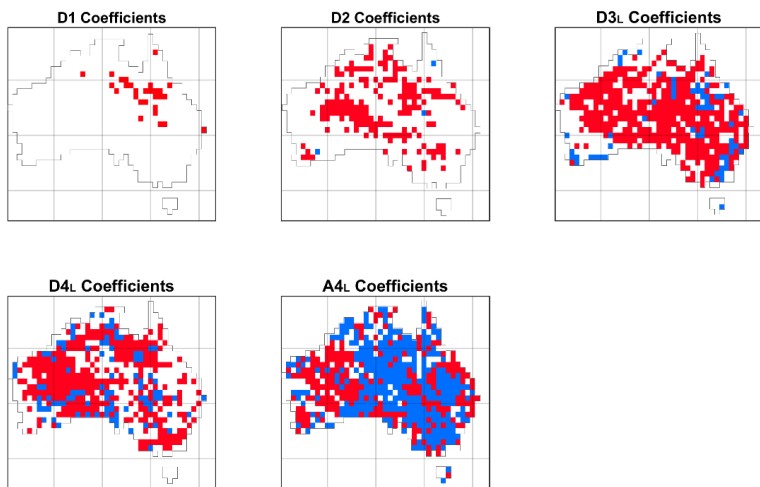

5   **Figure 4:** Coefficients for each decomposition level. For D1 and D2 no lags are used, red represents for these a positive coefficient and blue represents a negative coefficient. For $D3_L$, $D4_L$ and $A4_L$ (which include lags), red represents cells where all coefficients are positive. Blue represents cells where at least one lag had a negative coefficient.

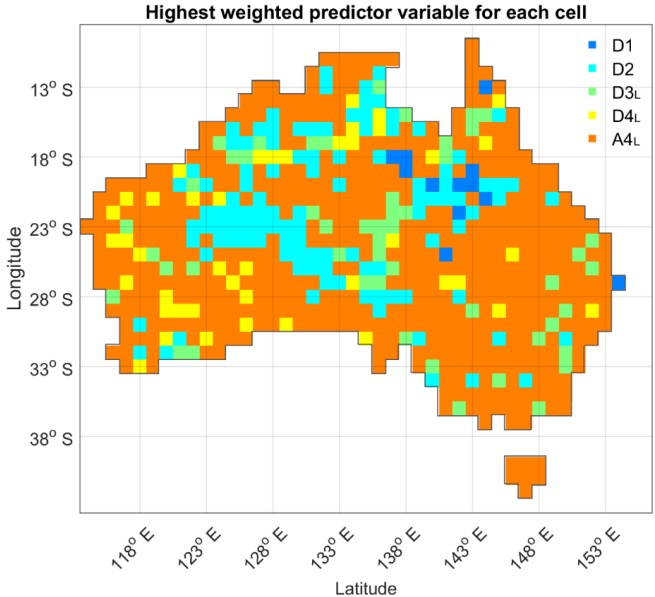

10   **Figure 5:** The variable with the highest relative weight in the regression for each cell across Australia. A4 is most dominant, however D2 is prominent in distinct areas throughout central Australia. D1, $D3_L$ and $D4_L$ all occur but with little spatial coherence.





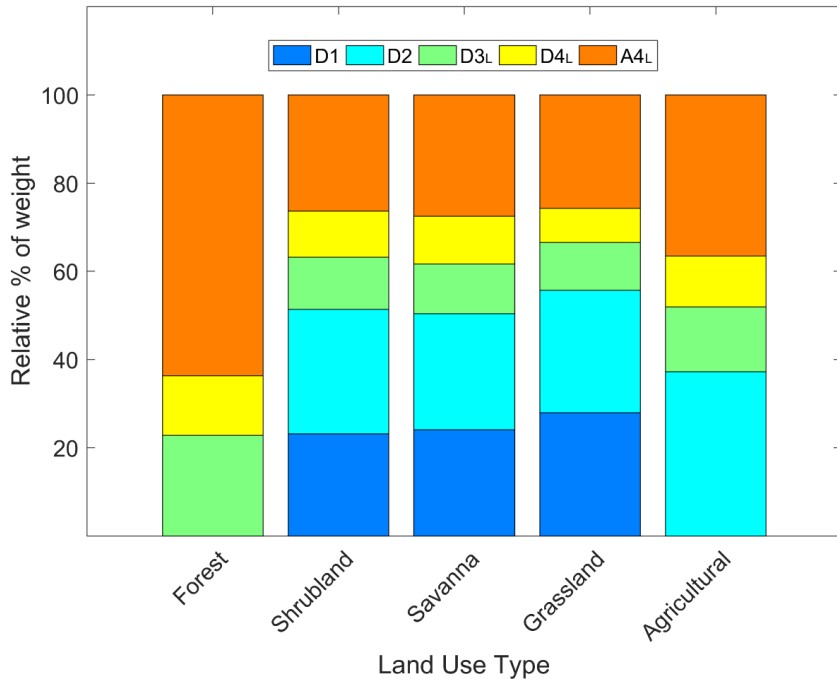

**Figure 6:** The relative weight of each decomposed TWS* for each land use type. Forests are A4$_L$ dominated, shrublands, savannas and grasslands are very similar with relative equal weights of D1, D2 and A4$_L$, while agricultural land is dominated by D2 and A4$_L$.

**Tables**

*Table 1: Subcategories of land use types as defined by MODIS*

| MODIS Land Use Type | Classification in this study |
|---|---|
| Evergreen needle leaf forest<br>Evergreen broad leaf forest<br>Deciduous needle leaf forest<br>Deciduous broad leaf forest | *Forest* |
| Closed shrublands<br>Open shrublands | *Shrubland* |
| Woody savanas<br>Savanas | *Savana* |
| Grassland | *Grassland* |
| Cropland<br>Cropland/Natural vegetation mosaic | *Agricultural land* |
| Barren | *Barren* |





**Appendix A**

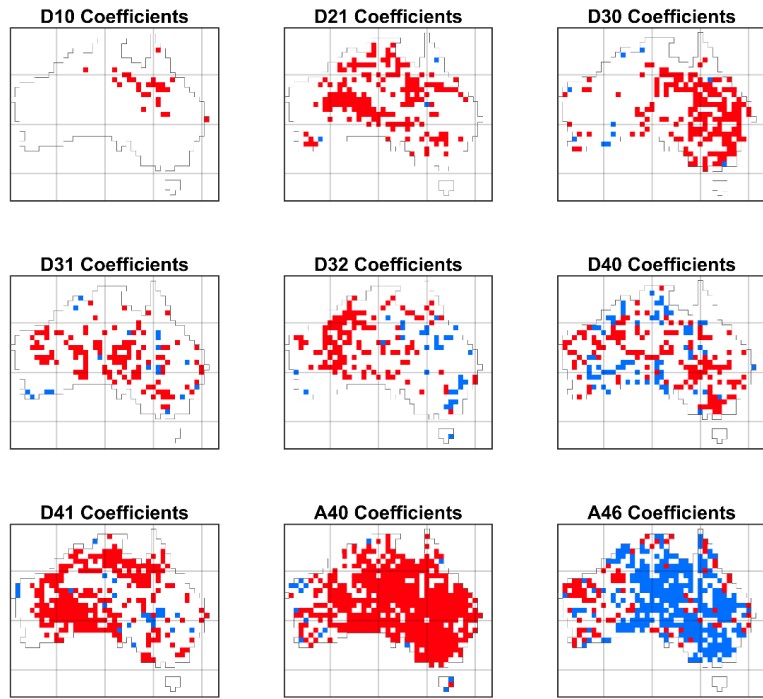

5    Figure A1: Coefficients for all 9 decomposition levels including lags. Red represents a positive coefficient and blue represents a negative coefficient.