# Peer review of "Large-scale vegetation responses to terrestrial moisture storage changes"

_Hydrology and Earth System Sciences, 2016_

## Referee Comment (RC1) · Anonymous Referee #1 · 7 Dec 2016

General comments: This is an interesting study, which extends a previous analysis by Yang et al. (2014) to examine how different-frequency components of the GRACE signal affect the temporal changes of surface greenness. It is packed with information, concepts, and unifying principles that are of great potential value in water and carbon cycle science. I am generally supportive of publication. However, I have one major concern that is needed to be addressed by the authors.

Major concern: It reads to me that the authors used a combination of different-frequency components of the GRACE data to correlate with monthly NDVI anomaly. I still do not understand what is the scientific and physical basis of comparing monthly NDVI with water storage components at a longer time scale, for example, D1 for 2 months and D4 for 16 months.

Additional minor comments: 1. I would add a figure showing different components of the GRACE signal (i.e., A1-A4 and D1-D4). This will give readers a more intuitive understanding of these components. 2. Section 2.3 Please specify which year of the MODIS Land cover data was used? 3. Page 7 Line 5. Should be TWS* data is better associated. Delete "the". 4. I am wondering if the higher-frequency GRACE components (i.e., level-1 and level-2) are more related with variations of soil moisture within shallower soil zones, and lower-frequency ones are more related with moisture changes within deep zones or groundwater. If this were the case, it could somehow answer my major concern.

---

## Referee Comment (RC2) · Anonymous Referee #2 · 1 Jan 2017

Major Comments

- GRACE data do not have a sensitivity to a spatial resolution of 1-degree. The authors even divided Tasmania into three different land use types. I highly doubt there is any meaningful information from GRACE at the spatial scale down to 100 km.

- This manuscript is difficult to assess partly because it does not provide a detail explanation of their approach. I do not understand Section 3. Temporal variations of vegetation cover (NDVI) and total water storage (GRACE) are dominantly at a seasonal frequency. The authors removed such largest variability in the data and examine only the residual data after removing climatology based on monthly data over many years. I do not understand the rationale of analyzing only the secondary signals (the residuals) to study vegetation response to terrestrial water storage.

[Figure]

- I am surprised that I do not see any time-series plot in their analyses. Also, I do understand what various time-scales indicated in Section 3.1 imply.

- Technical advance seems to be moderate (but, again, its validity is difficult to judge due to lack of sufficient explanation in Section 3)

- Science quality is low to moderate. I am not convinced that this manuscript contains sufficient science advance or discovery that warrants publication in the journal HESS. Discussion in the last paragraph of Section 4 and Section 5 seem to be trivial and just descriptive without any quantification.

---

## Author Comment (AC1) · 19 Jan 2017

Dear Reviewer #1.

Thank you for taking the time to read our manuscript and to provide valuable comments. We are glad you found the work interesting and are supportive of its publication.

We would hereby like to address your major concern by explaining a bit more the scientific and physical basis of the comparison. The core idea in the paper is that high frequency signals (D1, D2, A1, and A2) correspond to storage changes of shallow soil moisture, while low frequency signals such as D3, D4, A3 and A4 correspond to deeper soil moisture or even groundwater. We previously proved this in another manuscript which is currently under review (revised version) for Journal of Hydrology. This is why forests mostly correspond with low frequency changes; their roots are

accessing water storage that is changing at a lower frequency (though it is entirely possible for high frequency signals to also exist in forests and low frequency signals to exist in grasslands etc.). Land covers (hence NDVI) dominated with shrubs and grasses (shallow roots) correspond to higher frequency signals where moisture change is more dynamic. This might not have been as clear as we intended in the manuscript, we can make changes to make this clearer in future revisions.

Additional minor comments:

1. Such a figure as you suggest is in our other manuscript which is in review (as mentioned above). We agree that it would be useful too in this paper, and will include an alternative version of it in the main text of future revisions. See figure 1.

Figure 1: An example of a wavelet decomposition from a cell in central South Australia (29°S 136°E). Notice the visible trends in the approximations, which are normalised in the details.

2. The MODIS land use type data is from 2012, this information will included in future revisions.

3. Thank you, this will be included in future revisions.

4. Yes, what you have commented is absolutely correct. I hope this has been answered in our response to your major concern. We will try to make this clearer in revised versions.

[Figure]

[Figure]

**Fig. 1.**

---

## Author Comment (AC2) · 19 Jan 2017

Dear Reviewer #2.

Thank you for taking the time to read our manuscript and to provide your valuable comments.

Major comments:

- We agree that the 100km x 100km is not the true GRACE resolution, however gridded GRACE data is provided by NASA at a resolution of roughly 100km x 100km (1 degree) and freely accessible online (http://grace.jpl.nasa.gov/data/get-data/). So long as an appropriate scaling function has been applied to the gridded it is considered accurate, as extensively discussed in Landerer and Swenson (2012) and we have applied

such a scailing function as supplied by NASA. Gridded GRACE data are used in most publications concerning GRACE such as in Yang et al. (2014), Feng et al. (2013), Tajdarul et al. (2008), Mulder et al, (2015) and Becker et al. (2011).

Tasmania is split into 3 different land use types based on the MODIS data set (MDC12Q1). Hence, we are confident that it is reasonable to use this GRACE product in combination with the MODIS data set for categorising land use types.

- For variables with strong seasonality, a statistical relationship between them does not necessarily mean a physical relationship exists. For example, temperature and precipitation of an area, without removing the seasonality, may have very strong correlation, but physically they are both a result of Earth's revolution of the Sun. They do not necessarily have a direct physical relationship with one another. Thus, to examine the physical relationship between seasonal variables, it is a common approach to remove seasonality before the correlation analysis, as found in many previous studies such as Yang et al. (2014), Zaitchik et al. (2008), Crowley et al. (2006). Specifically for vegetation, its condition is likely related to soil moisture, but may also be influenced by temperature and solar radiation. Without removing the seasonality, the statistical relationship between NDVI and TWS may include the influences of other seasonal variables (e.g., solar radiation and temperature).

The decomposition of the GRACE data allows utilisation of GRACE information at different temporal frequencies. Generally, low frequency signals should correspond to deeper rooted vegetation where moisture changes are less dynamic and higher frequency signals should correspond to shallower rooted vegetation where moisture changes are more dynamic (though it is entirely possible for high frequency signals to also exist in forests and low frequency signals to exist in grasslands etc). We use this method to reveal the moisture dependence of different vegetation types at these temporal frequencies, this is a new and innovative method and has not been used for explaining NDVI changes. In the revised version (section 3 methodology) we emphasise the anomaly calculation and its relevance.
- Because we analyse so many cells it is not practical to show time series of data. However, in the revised version we add in a time series of one cell to help further demonstrate the concept. An example of this inclusion is shown below (figure 1 - the actual figure will appear on the last page of this letter)

Figure 1. An example of the time series from a single cell. The new estimate uses the coefficients from A40, A46 and D4 as chosen by the stepwise regression. Pearsons coefficient (r) between the decomposed GRACE estimate and NDVI* is 0.872, compared with 0.665 when using raw GRACE TWS*.

- We are confident that the further explanation of the method in the paper and in the replies to the comments above shows the validity of the method.

- In this manuscript we show (1) An advancement in understanding the moisture dependence of vegetation types in Australia and (2) The application of a novel method expanding the use of GRACE data, both innovative and making contributions to the wider scientific community. This study builds on previous works that have linked NDVI with precipitation, soil moisture and GRACE (Chen et al., 2014, Huxman, 2004, Méndez-Barroso et al, 2009, Wang et al., 2007, Yang et al., 2014).

Yang et al. (2014) is the first study using GRACE TWS to examine large-scale moisture storage effect on terrestrial vegetation performance. It has been cited by papers published in Review of Geophysics, JGR, Remote Sensing of Environment, Journal of Hydrology and Environmental Research Letters. However, part of GRACE TWS is beyond the reach of root zones, which is irrelevant to vegetation functioning. In our manuscript, we address this issue by proposing a method to use more shallow water storage signals in GRACE data. We clearly make an additional scientific advance by way of above points 1 and 2. The last paragraph in section 5 further discusses specific results. We have revised section 4 and 5 and taken care that the significance is more clearly stated, given that we used a new method we believe that the results are not 'trivial', rather, they support our hypothesis. The last paragraph of section 5 highlights

the application and contribution of the study presented.

The significance of this study can be also seen from the number of views and downloads having occurred to the discussion paper. As of Janurary 19th 2017, this discussion paper was downloaded 251 times compared to 101, 126, and 185 times for three other papers (with subjects in ecohydrology, and remote sensing and GIS) published within the similar time frame in HESS for discussion.

References:

Becker, M., Meyssignac, B., Xavier, L., Cazenave, A., and Decharme, B., 2011. Past terrestrial water storage (1980–2008) in the Amazon Basin reconstructed from GRACE and in situ river gauging data. Hydrology and Earth System Sciences, 15, 533-546.

Chen, T., R.A.M, d. J., Lui, Y., Van der Werf, G. R., and Dolman, A. J., 2014. Using satellite based soilmoisture to quantify the water driven variability. Remote Sensing of Environment, 140, 330-338.

Crowley, J., Mitrovica, J., Bailey, R., Tamisiea, M., and Davis, J., 2006. Land water storage within the Congo Basin inferred from GRACE satellite gravity data. Geophysical Research Letters, 33, doi: 10.1029/2006GL027070

Feng, W., Zhong, M., Lemoine, J.-M., Biancale, R., Hsu, H.-T., and Xia, J., 2013. Evaluation of groundwater depletion in North China using the Gravity Recovery and Climate Experiment (GRACE) data and ground-based measurements. Water Resources Research, 49, 2110-2118.

Huxman, T.E., Smith, M.D., Fay, P.A., Knapp, A.K., Shaw, M.R., Loik, M.E., Smith, S.D., Tissue, D.T., Zak, J.C., Weltzin, J.F. and Pockman, W.T., 2004. Convergence across biomes to a common rain-use efficiency. Nature, 429, 651-654.

Méndez-Barroso, L. A., Vivoni, E. R., Watts, C. J., and Rodríguez, J. C., 2009. Seasonal and interannual relations between precipitation, surface soil moisture and vegetation dynamics in the North American monsoon region. Journal of Hydrology, 377,

59-70.

Mulder, G., Olsthoorn, T., Al-Manmi, D., Schrama, E., and Smidt, E., 2015. Identifying water mass depletion in northern Iraq observed by GRACE. Hydrology and Earth System Sciences, 19, 1487-1500.

Tajdarul, S., Famiglietti, J., Rodell, M., Chen, J., and Wilson, C., 2008. Analysis of terrestrial water storage changes from GRACE and GLDAS. Water Resources Research, 44, W02433, doi:10.1029/2006WR005779.

Wang, X. W., Xie, H. J., Guan, H. D., and Zhou, X. B., 2007. Different responses of MODIS-derived NDVI to root-zone soil moisture in semi-arid and humid regions. Journal of Hydrology, 340, 12-24.

Yang, Y., Long, D., Guan, H., Scanlon, B. R., Simmons, C. T., Jaing, L., and Xu, X., 2014. GRACE satellite observed hydrological controls on interannual and seasonal variability in surface greenness over mainland Australia. Journal of Geophysical Research: Biogeosciences, 119, 2245-2260.

Zaitchik, B., Rodell, M., and Reichle, R., 2008. Assimilation of GRACE terrestrial water storage data into a land surface model: Results for the Mississippi River basin. Journal of Hydrometeorology, 9, 535-548.

[Figure]

**Fig. 1.**

---

## Author Response (AR1)

[revised manuscript text omitted]

Comment [T1]: Added word

Comment [T2]: This has been added to clarify the nature of the scaling function applied

[revised manuscript text omitted]

**Comment [T4]:** This section has been added to the methodology. Similar text previously appeared in section 2 but has been moved to here and modified as we feel it fits in the 'methodology' section better than 'data'

**Comment [T5]:** Addition of figure as suggested by reviewer 1 minor comment 1 (See actual figure at end of document.) Note that figure numbers have been updated accordingly

**Comment [T6]:** An additional sentence to clarify using the decomposed time series' in a stepwise regression

[revised manuscript text omitted]

Evergreen broad leaf forest

Deciduous needle leaf forest

Deciduous broad leaf forest | *Forest* |
| Closed shrublands

Open shrublands | *Shrubland* |
| Woody savanas

Savanas | *Savana* |
| Grassland | *Grassland* |

| Cropland | Agricultural land |
|---|---|
| Cropland/Natural vegetation mosaic | |
| Barren | *Barren* |

**Appendix A**

[Figure]

Figure A1: Coefficients for all 9 decomposition levels including lags. Red represents a positive coefficient and blue represents a negative coefficient.

Dear Editor and reviewers,

Thank you for taking the time to provide valuable feedback towards our manuscript entitled 'Large-scale vegetation responses to terrestrial moisture storage changes'.

In this response document we provide details on the changes which have been made to the manuscript in response to each comment from the editor and each reviewer, or rebuttal to comments where appropriate. A revised version of the manuscript with additions/changes marked up as they have been made is included below.

**Response to comments from the Editor**

*Dear authors,*
*Thank you for your submission to HESS. We have received two sets of referee's comments. Reviewer #2 is more critical. He is particularly concerned about the lack of clarity in the methodological approach, and mentions that this lack of clarity makes the discussion and interpretation of result difficult and therefore affects the value of the contribution. Reviewer #1 is more positive but has also listed some major concerns.*

*I agree with the reviewers that there are some key aspects of the methodology that are not clear in the paper. The authors mention, in the responses to the reviewers, that they will incorporate more details in the revised paper, which might hopefully address the reviewer's concerns.*

We have made an effort to make the methodology more clear (Section 3). In particular we have added 2 extra figures as suggested by the reviewers (figures 2 and 5).

*The authors mention a paper under review, in response to the comments to reviewer #1. Please note that the paper methods and discussion cannot be based on unpublished methods/results. This paper needs to include all relevant information for the revision process, and it cannot be accepted if a relevant portion of the methodology depends or has been tested in a paper that is not available. The authors need to incorporate the information either in the main text or as supplementary material. It is also very important to notice that the results of the work presented in this manuscript should be relevant even if the other paper (now under review in journal of hydrology-JoH) is published.*

The paper mentioned is still under review with Journal of Hydrology. As such we have incorporated more information in the manuscript in place of referencing the other unpublished paper. To be clear, the paper under review with JoH uses a similar method by decomposing GRACE TWS data using wavelets. However, the focus of that paper is to use the wavelets to partition GRACE TWS data into different vertical components (i.e. shallow and deep soil moisture, groundwater etc). So while there is some overlap in the method, there is no overlap in aims, results of findings of this manuscript submitted to HESS.

*After the new manuscript is submitted addressing all the referees' concerns, it will be send back to the reviewers for further assessment of the revised manuscript.*

**Response to comments from Reviewer #1**

*General comments: This is an interesting study, which extends a previous analysis by Yang et al. (2014) to examine how different-frequency components of the GRACE signal affect the temporal changes of surface greenness. It is packed with information, concepts, and unifying principles that are of great potential value in water and carbon cycle science. I am generally supportive of publication. However, I have one major concern that is needed to be addressed by the authors.*

*Major concern: It reads to me that the authors used a combination of different frequency components of the GRACE data to correlate with monthly NDVI anomaly. I still do not understand what is the scientific and physical basis of comparing monthly NDVI with water storage components at a longer time scale, for example, D1 for 2 months and D4 for 16 months.*

We would hereby like to address your major concern by explaining better the scientific and physical basis of the comparison. The core idea in the paper is that high frequency signals (D1, D2, A1, and A2) correspond to storage changes of shallow soil moisture, while low frequency signals such as D3, D4, A3 and A4 correspond to deeper soil moisture or even groundwater. We previously proved this in another manuscript which is currently under review (revised version) for Journal of Hydrology. This is why forests mostly correspond with low frequency changes; their roots are accessing water storage that is changing at a lower frequency. Land covers (hence NDVI) dominated with shrubs and grasses (shallow roots) correspond to higher frequency signals where moisture change is more dynamic.

A condensed excerpt of the above explanation has been added to the manuscript, end of section 3.3.

*Additional minor comments: 1. I would add a figure showing different components of the GRACE signal (i.e., A1-A4 and D1-D4). This will give readers a more intuitive understanding of these components.*

A figure such as you have suggested has been added and is now fig. 2.

*2. Section 2.3 Please specify which year of the MODIS Land cover data was used?*

MODIS land cover data from 2012 was used, this has now been specified in section 2.3 of the manuscript

*3. Page 7 Line 5. Should be TWS\* data is better associated. Delete "the".*

Addressed

*4. I am wondering if the higher-frequency GRACE components (i.e., level-1 and level-2) are more related with variations of soil moisture within shallower soil zones, and lower-frequency ones are more related with moisture changes within deep zones or*

Yes, what you have commented is absolutely correct. I hope this has been answered in our response to your major concern. We hope the additions to the methodology section of the manuscript have made this clearer.

**Response to comments from Reviewer #2**

*Major Comments*
*1. GRACE data do not have a sensitivity to a spatial resolution of 1-degree. The authors even divided Tasmania into three different land use types. I highly doubt there is any meaningful information from GRACE at the spatial scale down to 100 km.*

**REBUTTLE:** We agree that the 100 km x 100 km is not the true GRACE resolution. The true resolution is much coarser than 100km x 100km, closer to 300 km. The 100 km resolution stems from the original GRACE measurements, and the subsequent truncation and Gaussian averaging filters which are applied to produce TWS. In addition to the resolution loss, the algorithms applied to generate TWS also introduce "leakage" error (Landerer and Swenson 2012).

Gridded gain factors have been produced based on land surface modelling (e.g., GLDAS-NOAH) aiming to correct the leakage error. This gain factors have been made in the 1 by 1 degree grids. Thus, the gridded GRACE TWS products are provided in the same apparent resolution.

Although it is not the true resolution of GRACE, Landerer and Swenson (2012) state that "The gain factors derived here are based on simulated TWS variations, and are independent of the actual GRACE observations. Their purpose is to extrapolate the GRACE data to finer spatial scales that are not well resolved by the current GRACE satellites. It is important to keep in mind that while these fine scales are not truly measured by GRACE, our gridded-TWS estimates represent these scales to the degree to which a scaling relationship can recover them. This scaling relationship also enables us to quantify leakage and measurement errors based on signal patterns of TWS". This further supports the use of such gridded 1 degree TWS data.

It is not uncommon to use the higher resolution gridded data when the relationships of two datasets of different spatial resolutions are investigated. In our case, we have NDVI of higher resolution (0.25 degree), and GRACE of lower resolution. We decide to use this gridded GRACE data because it is better than simply applying a disaggregation of GRACE data of original spatial resolution. Alternatively, we could aggregate NDVI data to match the original GRACE resolution, but this would lose NDVI information in investigating the relationship.

A similar approach to what we adopted has been used in other published studies. For example, Yang et al. (2014) used gridded 1 degree GRACE data to investigate vegetation - TWS relationship. Becker et al. (2011), using the gridded 1 degree GRACE data, investigated the relationship between TWS and point measurements of water levels in Amazon River. Yi & Wen (2016) used the 1 degree GRACE TWS to develop a drought index for America.

Tasmania is split into 3 different land use types based on the MODIS data set (MDC12Q1).The high resolution MODIS NDVI data is degraded to 1 degree resolution to match the 1degree GRACE TWS grids.

In summary, we conclude that it is appropriate to use the 1 degree GRACE TWS product to investigate the dependence of NDVI on large-scale water storage in this study.

*2.This manuscript is difficult to assess partly because it does not provide a detail explanation of their approach. I do not understand Section 3. Temporal variations of vegetation cover (NDVI) and total water storage (GRACE) are dominantly at a seasonal frequency. The authors removed such largest variability in the data and examine only the residual data after removing climatology based on monthly data over many years. I do not understand the rationale of analyzing only the secondary signals (the residuals) to study vegetation response to terrestrial water storage.*

Further details have been added to section 3, we hope this, along with the 2 extra figures makes the approach clearer (See mark-ups in section 3 of manuscript.

**REBUTTLE/JUSTIFICATION of using anomalous data.**
For variables with strong seasonality, a statistical relationship between them does not necessarily mean a physical relationship exists. For example, temperature and precipitation of an area, without removing the seasonality, may have very strong correlation, but physically they are both a result of Earth's revolution around the Sun. They do not necessarily have a direct physical relationship with one another.

Thus, to examine the physical relationship between seasonal variables, it is a common approach to remove seasonality before the correlation analysis, as applied in many previous studies such as Yang et al. (2014), Zaitchik et al. (2008), Crowley et al. (2006).

Specifically for vegetation, its condition is likely related to soil moisture, but may also be influenced by temperature and solar radiation. Without removing the seasonality, the statistical relationship between NDVI and TWS may include the influences of other seasonal variables (e.g., solar radiation and temperature).

The decomposition of the GRACE data allows utilisation of GRACE information at different temporal frequencies. Generally, low frequency signals should correspond to deeper rooted vegetation where moisture changes are less dynamic and higher frequency signals should correspond to shallower rooted vegetation where moisture changes are more dynamic (though it is entirely possible for high frequency signals to also exist in forests and low frequency signals to exist in grasslands etc). We use this method to reveal the moisture dependence of different vegetation types at these temporal frequencies, this is a new and innovative method and has not been used for explaining NDVI changes. In the revised version (section 3 methodology) we emphasise the anomaly calculation and its relevance.

*3. I am surprised that I do not see any time-series plot in their analyses. Also, I do understand what various time-scales indicated in Section 3.1 imply.*

Because we analyse so many cells it is not practical to show time series of data. However, we have added a time series of one cell to help further demonstrate the concept. This now appears as figure 5.

*4. Technical advance seems to be moderate (but, again, its validity is difficult to judge due to lack of sufficient explanation in Section 3)*

We are confident that the further explanation of the method in the paper and in the replies to the comments above shows the validity of the method.

*5. - Science quality is low to moderate. I am not convinced that this manuscript contains sufficient science advance or discovery that warrants publication in the journal HESS. Discussion in the last paragraph of Section 4 and Section 5 seem to be trivial and just descriptive without any quantification.*

**REBUTTLE:** In this manuscript we show (1) an advancement in understanding the moisture dependence of vegetation types in Australia; and (2) the application of a novel method expanding the use of GRACE data, this is the first time this method is applied to analyse NDVI data. Both are innovative and making contributions to the wider scientific community. This study builds on previous works that have linked NDVI with precipitation, soil moisture and GRACE (Chen et al., 2014, Huxman, 2004, Méndez-Barroso et al, 2009, Wang et al., 2007, Yang et al., 2014).

Yang et al. (2014) is the first study using GRACE TWS to examine large-scale moisture storage effect on terrestrial vegetation performance. It has been cited by papers published in Review of Geophysics, Journal of Geophysical Research, Remote Sensing of Environment, Journal of Hydrology, and Environmental Research Letters. However, part of GRACE TWS is beyond the reach of root zones, which is irrelevant to vegetation functioning. In our manuscript, we address this issue by proposing a method to use the more shallow water storage signals of GRACE data. We clearly make an additional scientific advance by way of above points 1 and 2. The last paragraph in section 5 further discusses specific results. We have revised section 4 and 5 and taken care that the significance is more clearly stated, given that we used a new method to better reveal the large-scale biomass production (reflected in NDVI) and moisture storage (reflected in TWS), we believe that the results are not 'trivial'. The last paragraph of section 5 highlights the application and contribution of the study presented.

The significance of this study can be also seen from the number of views and downloads having occurred to the discussion paper. As of January 19[th] 2017, this discussion paper was downloaded 251 times compared to 101, 126, and 185 times for three other papers (with subjects in ecohydrology, and remote sensing and GIS) published within the similar time frame in HESS for discussion.

[revised manuscript text omitted]

Comment [T1]: Added word

Comment [T2]: This has been added to clarify the nature of the scaling function applied

[revised manuscript text omitted]

**Comment [T4]:** This section has been added to the methodology. Similar text previously appeared in section 2 but has been moved to here and modified as we feel it fits in the 'methodology' section better than 'data'

**Comment [T5]:** Addition of figure as suggested by reviewer 1 minor comment 1 (See actual figure at end of document.) Note that figure numbers have been updated accordingly

**Comment [T6]:** An additional sentence to clarify using the decomposed time series' in a stepwise regression

[revised manuscript text omitted]

 Evergreen broad leaf forest
 Deciduous needle leaf forest
 Deciduous broad leaf forest | *Forest* |
| Closed shrublands
 Open shrublands | *Shrubland* |
| Woody savanas
 Savanas | *Savana* |
| Grassland | *Grassland* |

| Cropland | Agricultural land |
|---|---|
| Cropland/Natural vegetation mosaic | |
| Barren | *Barren* |

**Appendix A**

[Figure]

Figure A1: Coefficients for all 9 decomposition levels including lags. Red represents a positive coefficient and blue represents a negative coefficient.